# *SlMYC2* Mediates the JA Pathway by Responding to Chlorocholine Chloride in the Regulation of Resistance to TYLCD

**DOI:** 10.3390/plants14091353

**Published:** 2025-04-30

**Authors:** Yanan Ma, Liangfang Wang, Zuozeng Cao, Hui Wang, Fu Wang, Wenying Zhu

**Affiliations:** 1College of Horticulture, Qingdao Agricultural University, Qingdao 266109, China; 2Engineering Laboratory of Genetic Improvement Horticultural Crops of Shandong Province, Qingdao 266109, China

**Keywords:** tomato, CCC, yellow leaf curl virus, *SlMYC2*, jasmonic acid

## Abstract

Tomato yellow leaf curl disease (TYLCD) significantly affects tomato yield. The jasmonic acid (JA) pathway is crucial in the defence response of plants; however, its role in plant resistance to TYLCD remains undefined. In production, CCC (chlorocholine chloride) is often used to cultivate strong seedlings to enhance seedling vitality and improve stress resistance. However, the mechanism through which CCC enhances disease resistance in tomatoes remains unclear. In this study, tomato seedlings were exogenously sprayed with 300 mg/L CCC before and after inoculation with tomato yellow leaf curl virus (TYLCV). The results indicated that no significant tomato yellow virus disease phenotype was observed in tomato seedlings after spraying with CCC and subsequent inoculation with the virus. Spraying CCC on seedlings inoculated with the virus and exhibiting typical phenotypes can significantly alleviate the yellowing and curling symptoms of new leaves and improve photosynthesis-related indicators in tomato plants. The detection of virus copy numbers within the plants revealed that the virus copy numbers in plants treated with CCC were significantly lower than those in the control group. Transcriptomic analysis revealed that, after spraying CCC, the key enzyme genes *AOS2* and *AOC* in the JA synthesis pathway in tomatoes were significantly upregulated, whereas the expressions of *JAZ2* and *MYC2* genes, which negatively regulate JA synthesis, were significantly downregulated. In the stable state, JAZ proteins interact with *MYC2* and inhibit its transcriptional activity of *MYC2*. Tomatoes overexpressing *MYC2* and *JAZ2* exhibit a significant decrease in TYLCD resistance. These results indicated that exogenous spraying CCC affected the expression of genes such as *MYC2* and *JAZ2*, and then regulated JA pathway, increased the endogenous JA content in plants, and enhanced the disease resistance of tomato plants to TYLCD. This study provides a scientific reference for effectively preventing and controlling TYLCD in tomato production and reducing the influence of TYLCD on tomato yield and quality.

## 1. Introduction

Tomato (*Solanum lycopersicum* L.) is an annual herbaceous plant that belongs to the Solanaceae family and is native to South America; it is utilised both as a vegetable and fruit. It is highly esteemed for its abundant carotenoids and vitamins C and B, which significantly enhance its economic value. Tomato yellow leaf curl virus (TYLCV), a single-stranded circular DNA virus, belongs to the genus Begomovirus within the Geminiviridae family. This virus, transmitted by the whitefly (*Bemisia tabaci*), induces tomato yellow leaf curl disease (TYLCD) in various plants such as tomatoes, cotton, tobacco, watermelon, pepper, and gourds. Infected tomato plants display symptoms such as yellowing and curling of leaves, as well as stunted growth and wilted foliage, potentially leading to catastrophic crop failures. TYLCV is a seed-transmissible geminivirus with a seed infectivity rate that varies between 20% and 100% in tomato plants infected by whitefly-mediated transmission and agricultural inoculation. Despite numerous constraints, effective management strategies to control TYLCD are yet to be fully established in agricultural settings.

When plants encounter stress during their growth and development, a complex regulatory network is established within the plant to counteract external biotic or abiotic stresses. In this network, various plant hormones, including jasmonic acid (JA), abscisic acid (ABA), gibberellin (GA), and 1-aminocyclopropane-1-carboxylic acid (ACC), play a critical role. Plants have evolved complex and sophisticated defence response pathways controlled by plant hormones, such as JA and salicylic acid (SA), to counter the different invasion patterns of pathogens [1]. JA, a major immune hormone, promotes plant defence against mechanical damage, insect attacks, and pathogen infections and plays a significant regulatory role in various plant resistances [2].

After decades of research, previous studies have elucidated the main core signalling pathway of JA involvement in plant resistance in two model plants: Arabidopsis and tomato. This pathway comprises multiple interconnected protein–protein interaction modules that regulate the transcriptional state of hormone-responsive genes. In addition to plant hormones, many transcription factors (such as *MYC*, *AP2/ERF*, *WRKY*, *NAC*, etc.) play important roles in plant defence responses. Among the transcription factors induced by jasmonic acid that have been extensively studied, *MYC2* is one of the most important and most researched transcription factors. The main regulatory factor for jasmonic acid responses in various crops, including Arabidopsis, tomato, tobacco, primrose, banana, and rice, and it controls various aspects of multiple defence responses [3,4,5,6,7,8,9,10,11,12,13,14,15]. MYC transcription factors are widely present in both animals and plants, and as a subfamily of basic helix–loop–helix (bHLH) transcription factors, they participate in various stress resistance processes in plants and are important regulatory factors in the jasmonic acid signalling pathway. The JA signalling pathway in Arabidopsis thaliana has been clarified, in which JAZ protein acts as a transcription inhibitor, and *MYC2* regulates all aspects of JA reaction to varying degrees, and JAZ protein interacts with *MYC2* to inhibit the transcription activity of *MYC2* [3,4,5,6,7,16,17,18,19,20,21]. When the plant is under stress, both of them synergistically regulate JA signalling pathway and increase JA content to resist stress [2,6,22]. Moreover, it has been proved in periwinkle, banana, rice, and other crops that *MYC2* can mediate systemic resistance of plants by regulating JA signalling pathway [10,11,12,13,14,15].

Chlorocholine chloride, also known as 2-chloro-N,N,N-trimethyl-ethanaminium chloride (CCC), is a quaternary ammonium salt used as a plant growth regulator. CCC is predominantly utilised in the cultivation of horticultural crops to foster vigorous seedling development, although there is a significant lack of research on its impact on disease resistance in tomatoes. A study conducted by Sayanti suggested that CCC treatment improved the resistance of *Stevia rebaudiana* to leaf spot disease [23]. Furthermore, a study conducted by Zhang revealed that the application of CCC treatment to *Ginkgo biloba* notably elevated the concentrations of total polyphenols, flavonoids, and anthocyanins, as well as the activities of phenylalanine ammonia-lyase and chalcone isomerase [24]. This enhancement corresponds to improved resistance to leaf spot disease. However, to the best of our knowledge, there have been no reports on the use of CCC to bolster disease resistance in tomatoes.

There have been many reports on the regulation of the JA signalling pathway by *MYC2* to improve plant resistance, but the role of JA in tomato resistance to TYLCD is rarely reported. In addition, CCC, as a plant growth retardant, has been reported in the cultivation of strong seedlings, but there is no report on its improvement in plant resistance and molecular mechanism. Through the observation of our research team in the field in the early stage, it was found that the occurrence of TYLCD in tomatoes was significantly alleviated after CCC spraying, which could greatly reduce the impact of TYLCD on tomato fruit yield and quality. Therefore, the molecular mechanism of CCC improving tomato resistance to TYLCD was discussed in this study. In this study, we found that exogenous CCC induced the biosynthesis and signal transduction pathway of JA by suppressing the expression of *MYC2* in tomato, and that the activated JA signalling pathway predominantly engaged with the *JAZ2/MYC2* module and two pivotal transcription factors within the ethylene (ET) pathway, namely *ERF4* and *ERF5.* This interaction is concurrent with an elevation in JA levels and an enhancement in photosynthetic parameters, which leads to the restoration of leaf colour in tomatoes and enhances their resistance to yellow curl leaf disease. These findings provide a reliable approach for identifying specific exogenous inducers that can trigger enhanced resistance to yellow curl leaf disease in tomatoes.

## 2. Results

### 2.1. The Effect of Exogenous Application of CCC on Tomato Resistance to TYLCD

The application of the 3.1 CCC treatment significantly enhanced the resistance of tomatoes to yellow curl leaf disease. As shown in Figure 1a, CCC treatment markedly improved the plants’ resistance to this disease. The incidence rate of the treatment group sprayed CCC every five days after inoculation was significantly lower than that of the treatment group sprayed CCC after the disease occurs, and the treatment group inoculated with virus only, demonstrating the effective preventive capabilities of the treatment. Furthermore, we observed significant improvements in leaf colour and curling degree in plants subjected to the CCC treatment. As illustrated in Figure 1b, following the onset of the disease, the leaf colour of tomato plants gradually transitioned from yellow-green to green after CCC treatment. In addition to the chlorophyll content, we detected a significant increase in the net photosynthetic rate of tomato leaves treated with CCC (Figure 1c,d). These results suggest that the enhanced resistance of tomatoes to yellow curl leaf disease following CCC treatment may be closely linked to physiological pathways, such as photosynthesis and chlorophyll synthesis.

To further confirm the resistance mechanism induced by CCC in tomatoes, we used RT-qPCR to assess the copy number of TYLCV in the leaves of the different treatment groups. The results indicated that the viral load in Treatment Group One was consistently lower than that in the virus-only control group from days 15 to 30, and that subsequent CCC treatment after the onset of the disease significantly reduced the TYLCV copy number within the tomato plants (Figure 1e,f). These findings provide robust support for the application of CCC to the study of disease resistance in tomatoes.

### 2.2. GO Enrichment Analysis of CCC Treatment

To further investigate the mechanism by which CCC induces disease resistance in tomatoes as well as its potential regulatory biological functions and processes, we conducted transcriptomic analyses of tomato plants sampled at 0 h, 2 h, 4 h, 6 h, 8 h, 12 h, and 24 h post-CCC treatment.

We identified a significant enrichment of 203, 114, 169, 147, 379, and 358 GO terms across the six time-points following CCC treatment (*p* < 0.01). Notably, there were consistent GO terms across all six time-points, particularly those related to the defence response (GO:0006952), DNA-binding transcription factor activity (GO:0003700), response to stimulus (GO:0050896), transcription regulatory factor activity (GO:0140110), and response to stress (GO:0006950). To ascertain the differences in resistance induction efficiency between the CCC-treated and untreated plants, we assigned differentially expressed genes (DEGs) to the aforementioned biological processes. In the CCC-treated plants, the defence response (GO:0006952) was rapidly activated at 2 h and persisted until 24 h. Additionally, the number of DEGs associated with the response to stimulus-related terms continued to increase within the 12 h timeframe, revealing the most significant changes in the transcriptome (the sum of upregulated and downregulated transcripts) (Figure 2).

### 2.3. KEGG Enrichment Analysis of DEGs Processed by CCC

The KEGG database serves as a valuable resource for understanding higher-level functions of biological systems. To gain deeper insights into the systemic effects of CCC treatment, we conducted a KEGG pathway enrichment analysis of the differentially expressed genes (DEGs). CCC treatment resulted in a significant increase in 6, 8, 13, 14, 14, and 8 KEGG pathways across the six time-points (*p* < 0.01). We identified one DEG that was commonly upregulated and 37 DEGs that were downregulated at these time-points (Figure 3). Furthermore, we observed that the majority of DEGs were significantly enriched in pathways such as “Plant hormone signal transduction”, “Plant-pathogen interaction”, “MAPK signalling pathway-plant”, “Phenylpropanoid biosynthesis”, and “Phenylalanine metabolism”. These findings indicated that CCC treatment enhanced the defence capabilities of tomato plants.

### 2.4. Role of CCC in Promoting JA Biosynthesis and Its Signalling Pathway in Tomatoes

Multiple signalling pathways, including those involving SA, JA, and ET, are typically employed in studies investigating the resistance of crops to various exogenous inducers. To further explore the mechanism by which CCC induces resistance to yellow curl leaf disease in tomatoes, we analysed the transcriptome sequencing results and constructed Venn diagrams (Figure 4a,b) based on the differentially expressed genes at various time-points. From the 12 distinct analytical groups, we identified two genes with consistent expression patterns: *MYC2* and *JAZ2*. After synthesising these findings and reviewing the relevant literature, we preliminarily concluded that the *JAZ2-MYC2* interaction module could respond to CCC signals and enhance the defence capacity of tomatoes. Notably, the genes associated with JA biosynthesis, *AOC* and *AOS2*, were significantly upregulated (Figure 4c). Subsequently, we conducted real-time quantitative PCR to monitor the expression dynamics of the differentially expressed genes. The results indicated a sustained downregulation of *MYC2* and *JAZ2* within 24 h of CCC treatment (Figure 5a,b). Importantly, comparative analysis revealed that the JA levels in the CCC-treated samples were significantly higher than those in the mock-treated control group (Figure 5c). Therefore, we infer that CCC contributes to the enhancement of JA signalling and biosynthesis in tomato plants.

### 2.5. Defence Signals Induced by CCC Are Regulated Through MYC2

*MYC2* is a crucial component of the jasmonic acid (JA) signalling pathway that directly regulates the expression of various downstream genes. Given *MYC2*’s ability to activate JA signal transduction and promote associated responses, we developed a keen interest in the interplay between *MYC2-JAZ2* and ET signalling. To investigate this relationship, we performed yeast two-hybrid assays, which revealed an interaction between *MYC2* and *JAZ2* (Figure 5a). Compared to the negative control pGADT7 + PGBKT7-*MYC2*, the coding sequence (CDS) of *MYC2* effectively interacted with the CDS of *JAZ2*. Furthermore, we linked *MYC2* and *JAZ2* to the C-terminus-(CLuc) and N-termini (NLuc) of firefly luciferase, respectively, and performed transient expression assays on *Nicotiana benthamiana* leaves. We further validated the interaction between *MYC2* and *JAZ2* in planta using luciferase complementation imaging (LCI) (Figure 5b). To determine the subcellular localisation of *MYC2*, we introduced *MYC2* fusion plasmids into Agrobacterium for the transient transformation of tobacco leaves. As illustrated, the *MYC2*-GFP fusion protein was localised in the nucleus (Figure 5c). Based on the results of these experiments, we hypothesised that the *MYC2-JAZ2* interaction module may collectively respond to CCC signals, thereby activating additional downstream signalling pathways.

GO analysis of the differentially expressed genes (DEGs) regulated by CCC revealed that the majority of DEGs were predominantly enriched in pathways related to defence responses, stress responses, and transcription factor activity associated with DNA binding. Notably, we observed significant enrichment of the key ethylene signalling transcription factor, *ERF4*, among the downregulated DEGs across 30 pathways. Additionally, the upregulated DEGs were enriched in the transcription factor ERF5 within six pathways. This led us to speculate that CCC treatment may also activate ethylene-related signalling pathways. To validate this hypothesis, we investigated whether *MYC2* directly interacts with *ERF4* and *ERF5*. Yeast two-hybrid (Y2H) experiments indicated that the coding sequence (CDS) of *MYC2* interacted with the CDS of both *ERF4* and *ERF5*, in contrast to the negative control pGADT7 + PGBKT7-MYC2 (Figure 6).

Building on this foundation, we explored the biological significance of the interactions between *ERF4, ERF5* and *MYC2*. Given the crucial roles of *ERF4* and *ERF5* in plant defence responses, we hypothesised that CCC-induced ET signalling regulates the expression of *ERF4* and *ERF5* through *MYC2*, thereby enhancing plant resistance to pathogens.

### 2.6. Knockout and Overexpression of MYC2 Affect the Disease Resistance of Tomatoes

To validate this hypothesis, we designed a gene knockout vector (Figure 7a) to reduce the expression levels of *MYC2* using CRISPR/Cas9 technology and employed overexpression techniques to enhance *MYC2* expression levels. This resulted in the establishment of two knockout lines and two overexpression lines, which were subjected to detailed observations following pathogen exposure.

We detected the expression level of *MYC2* gene in transgenic lines. The expression level of *MYC2* in overexpressed lines was significantly higher than that in wild-type lines (Figure 7c), while the expression level of knockout lines was significantly lower than that in wild-type line (Figure 7b). After inoculation with infectious clones, the viral load in plants and the disease incidence of plants were compared, respectively. After inoculation with TYLCV, the copy number of viruses in knockout lines was significantly lower than that in wild type (Figure 7d), while the copy number of viruses in over-expressed lines was significantly higher than that in wild type line (Figure 7d). Phenotypic observation showed that the disease incidence of gene-edited strains was significantly lighter than that of wild type, and the disease incidence of strains overexpressing *MYC2* was the most serious (Figure 7e). This further corroborates that CCC-induced resistance to yellow curl virus in tomatoes is primarily regulated by *MYC2*.

## 3. Discussion

### 3.1. Exogenous Application of CCC Can Effectively Enhance Resistance to TYLCD

TYLCD, caused by the tomato yellow leaf curl virus, is a serious threat to tomato production worldwide. Currently, two methods are used to control this virus. One is breeding by infiltrating resistant genes or markers from wild-type tomatoes into cultivated varieties, but this is time-consuming and resistance is not stable. Currently, all discovered sources of tomatoes resistant to TYLCV are wild tomatoes, and no resistance/tolerance genes have been found in cultivated tomatoes. Therefore, breeding TYLCV-resistant tomato varieties involves the incorporation of resistance genes from wild materials. Wild tomato species reported to have genes resistant to TYLCV include *S. chmielewski*, *S. pimpinellifolium*, *S. glandulosum*, *S. lycopersicoides*, *S. habrochaites*, *S. chilense*, and *S. peruvianum*. These wild species have been used to breed TYLCV-resistant tomatoes. To date, five resistance genes (markers) have been identified: *Ty-1* [25,26],*Ty-2* [27,28], *Ty-3* [29,30], *Ty-4* [31], and *Ty-5* [32]. *Ty-1* and *Ty-3* are the ones that have been widely used to cultivate tomatoes. Another approach is to interfere with the viral gene sequence using RNAi technology to inhibit viral gene replication, which involves genetic modifications. Currently, this approach cannot be applied to production, and consumer acceptance of genetically modified tomatoes is limited [33,34].

Plant-induced resistance provides a new method for enhancing tolerance to TYLCV. Induced resistance is a phenomenon in which plants quickly and strongly respond to subsequent pathogen infections if they have been exposed to certain elicitors by biological or abiotic factors. Induced resistance mainly includes induced systemic resistance (ISR) and acquired systemic resistance (SAR). ISR is induced by plant growth-promoting bacteria (PGPB), which rely on the JA and ET signalling pathways, whereas SAR is induced by pathogen infection or chemicals, such as SA or its derivative benzothiadiazole (BTH). Shang et al. (2011) [34] reported that appropriate concentrations of JA combined with SA, which is green and safe, enhanced tobacco resistance to RNA viruses and effectively improved plant broad-spectrum resistance. This method is easy to implement, but the biggest limiting factor is the relatively high cost of JA.

In this study, we sprayed tomato seedlings with exogenous CCC before and after inoculation with TYLCV and found that spraying CCC after inoculation with the TY virus significantly reduced the incidence and disease index of tomato seedlings. Spraying CCC onto tomato seedlings after disease onset significantly alleviated the viral disease phenotype in new leaves, improved photosynthesis-related indicators, and restored growth. Through the detection of endogenous hormones, we found that the application of CCC significantly increased JA content in tomato plants.

Based on the above results, we speculate that CCC induced tomato resistance to TYLCD may be realised through the following two ways: ① in previous studies, CCC was often used to cultivate strong seedlings in industrial seedling raising, indicating that although CCC delayed the growth of plants to a certain extent, the photosynthetic related indexes, chlorophyll content and stem diameter of plants showed an increasing trend, thus improving the resistance of plants to external adversity; ② we found that after spraying CCC, the level of JA in tomato could be increased, and JA was one of the important signals of ISR-mediated pathway. Therefore, we speculated that CCC might improve tomato’s resistance to TYLCD through ISR pathway. Our research shows that, exogenous application of CCC can be used as a convenient and effective method to enhance tomato resistance to TYLCD through non-transgenic and low-cost methods. We detected CCC residues in tomato fruits and seeds after spraying CCC. If the residues are lower than the threshold of international standards, the method of enhancing tomato resistance to TYLCD by spraying CCC externally has a wide application prospect in future tomato production.

### 3.2. SlMYC2 Mediates JA and ET Pathways by Responding to CCC in the Regulation of Resistance to TYLCD in Tomato

JA and ET, two major immune hormones, are involved in the response and regulation of various biotic and abiotic stresses in plants and are the two main signalling pathways of ISR. Do showed that *SlMYC2* acts as a master regulator play a major role in coordinating -mediated plant immunity [9]. Zhang confirmed that the lncRNA20718-miR6022-RLPs module induces the accumulation of reactive oxygen species (ROS) and reduces the content of jasmonic acid and ethylene, thereby inhibiting the resistance of tomatoes to late blight [35]. However, it is not clear whether the reaction process of tomato resistance to TYLCD is mediated by JA or ET metabolic pathway.

In the present study, we found that the exogenous application of CCC significantly enhanced the resistance of tomato seedlings to TYLCD. Through phenotypic analysis, viral copy number analysis, transcriptomic analysis, hormone determination, yeast two-hybrid, LCI, and transgenic validation experiments, the following conclusions were drawn. First, the exogenous application of CCC can significantly reduce the incidence and disease index of tomatoes, inhibit the replication of viruses within tomato plants, increase the resistance of tomatoes to TYLCD, and is accompanied by a significant increase in the content of jasmonic acid in tomato plants. Second, transcriptome analysis results showed that after the exogenous application of CCC, the expression levels of the jasmonic acid synthesis-related genes *MYC2* and *JAZ2* were significantly downregulated, whereas the key enzyme genes *AOS2* and *AOC* in the jasmonic acid synthesis pathway were significantly upregulated. The yeast two-hybrid results showed that there was an interaction between JAZ2 and MYC2, which is consistent with the results of previous studies. However, previous studies indicated that *JAZ2* is located upstream of *MYC2* and inhibits its transcriptional activity of *MYC2*. In this study, after spraying with CCC, both *JAZ2* and *MYC2* expression levels showed a decreasing trend, which is slightly different from the results of previous studies. We hypothesised that *MYC2* may be a gene that directly responds to CCC in tomato plants and that spraying CCC directly leads to a decrease in the expression level of *MYC2* in tomatoes. Furthermore, after the exogenous application of CCC, it is possible that the inhibitory effect of *JAZ2* on *MYC2* is relieved, leading to a simultaneous decrease in the expression levels of *JAZ2* and *MYC2* in tomatoes. Since both *JAZ2* and *MYC2* are negative regulators of JA synthesis, the simultaneous downregulation of them resulted in an increase in JA content in the plant. The increase in JA content enhances the systemic resistance of tomatoes, thereby inhibiting the replication of viruses within the plant. (Figure 8). Third, the gene *ERF4*, involved in ethylene pathway regulation, was downregulated, whereas *ERF5* was significantly upregulated, and there was an interaction between *MYC2* and both *ERF4* and *ERF5*, which, in turn, affected ethylene biosynthesis and resistance of plants to diseases (Figure 8). In addition, current research indicates that *ERF4* is a key JAZ interactor between ethylene and jasmonic acid hormone signalling pathways (Hu et al., 2022 [36]). Finally, tomatoes overexpressing *MYC2* exhibited a significant decrease in TYLCD resistance. The above conclusion verifies that *MYC2* mediates the regulation of tomato resistance to TYLCD through the jasmonic acid and ethylene pathways in response to CCC.

## 4. Materials and Methods

### 4.1. Growth Conditions and Treatment of Experimental Materials

The cultivar tomato ‘M82’, which is susceptible to yellow leaf curl disease, was utilised as the experimental material. Tomato plants were cultivated under standard sunlight greenhouse conditions at Qingdao Agricultural University in China. The seedlings were raised in a 72-hole tray. When the seedlings grew to three true leaves, they were transplanted into a flowerpot with a diameter of 25 cm for planting. After planting, the culture conditions were as follows: illumination period of 16 h/8 h (day/night), temperature of 25 °C/15 °C (day/night), and humidity of 75%. The common compound fertiliser was watered once a week with an irrigate. Plants with uniform growth, free from mechanical damage and pest infestations, were selected.

#### 4.1.1. Infectious Clone Inoculation

The inoculation of TYLCV was carried out by infectious cloning. The infectious cloning recombinant vector plasmids TY-1 and TY-2 of TYLCV were provided by Professor Bao Zhilong from the College of Horticulture Science and Engineering of Shandong Agricultural University.

Two days prior to inoculation, agrobacterium was activated by streaking, and single colonies were selected for shaking culture until an optical density (OD600) of approximately 0.6–0.8 was reached. On the day of inoculation, a buffer solution was prepared (comprising 10 mL of 0.1 M MES, 1 mL of 1 M MgCl_2_, 0.1 mL of 0.1 M As, with the remaining volume adjusted with distilled water). The buffer was then adjusted to an OD600 of 0.2, followed by a 1:1 mixture of TY-1 and TY-2, which was incubated at 28 °C with shaking for 2–3 h. Subsequently, Silwet L-77 was added at a ratio of 1:50, and the mixture was thoroughly homogenised before inoculation via vacuum infiltration. The inoculated seedlings were subjected to dark treatment overnight and then transferred to light conditions for cultivation the following day. The blank control group plants were inoculated with clean water.

#### 4.1.2. CCC (Chlorocholine Chloride) Treatment

CCC (chlorocholine chloride) is a quaternary ammonium salt plant growth regulator. The CCC used in this study was purchased from Shanghai Aladdin Reagent Co., Ltd. (Shanghai, China). The preparation method of the CCC solution is as follows: weigh 300 mg of choline chloride, adjust the volume to 1 L with ddH_2_O, prepare a 300 mg/L (about 0.002 mol/L) CCC solution, and put it in an opaque sprayer, which makes it ready for use.

The treatment was conducted by foliar spraying at a concentration of 300 mg/L until uniform water droplets were observed on the leaf surface. Several treatment methods were designed. ① For plants designated for RNA-seq, a single spray was applied, and leaf samples were collected from the same location 2 h post-application. ② For plants subjected to infectious clone inoculation, two protocols were established: one group received a spray every five days starting 24 h post-virus inoculation (C1), while another group was sprayed every five days following the onset of symptoms after virus inoculation (C2). Sampling was conducted prior to each subsequent spray, resulting in six applications. Fresh GRs were prepared for each experiment. In order to prevent the phenomenon that the local CCC concentration is too high due to the high temperature or strong light during the day and the rapid evaporation of water after spraying, the spraying time of CCC is from 17:00 to 18:00 every day. The experiment was divided into three treatment groups: Treatment One, Treatment Two, and a control group that received only the viral inoculation (C3).

### 4.2. Incidence Rate Statistics

A total of 35 tomato seedlings were established for each group. The proportion of symptomatic plants was recorded 21 and 27 days post virus inoculation, with the disease criterion defined as having three or more leaves exhibiting distinct symptoms associated with yellow leaf curl disease.

### 4.3. Measurement of Chlorophyll Content and Net Photosynthetic Rate

The chlorophyll content in tomato leaves was measured using a SPAD-502 PLUS chlorophyll metre at 21 and 27 days post-treatment. The net photosynthetic rate of the tomato leaves was assessed using a portable photosynthesis system (LI-6400XT, LI-COR Corporation, Lincoln, NV, USA). Five plants were selected for each treatment, and the leaves from the third to fifth nodes were measured on a clear day at 10:00 AM.

### 4.4. Detection of Viral Copy Number

#### 4.4.1. Standard Curve Construction

Following the method of Ding [37], plasmid standards were diluted in DNase/RNase-free water in a tenfold series to achieve final concentrations ranging from 5.3 × 10^8^ to 5.3 × 10^14^ copies/μL, resulting in seven plasmid samples used as templates. RT-qPCR was performed using a real-time fluorescence quantitative PCR instrument, with each concentration tested in triplicate.

#### 4.4.2. RNA Extraction and RT-qPCR

Leaf samples were homogenised in liquid nitrogen, and total RNA was extracted using RNAiso Plus (Takara Bio Inc., Dalian, China). cDNA was synthesised from the extracted RNA using the PrimeScript RT Master Mix (Takara). Quantitative PCR was performed using a LightCycler 480 real-time PCR detection system and 2×RealStar Fast SYBR qPCR Mix (GenStar Biology Science and Technology Co., Ltd., Beijing, China). The qPCR primers are listed in Appendix A. The qPCR reaction protocol was as follows: 95 °C for 2 min, followed by 40 cycles of 95 °C for 15 s, 56 °C for 30 s, and 72 °C for 30 s.

### 4.5. Determination of JA Content

The data acquisition system primarily consists of Ultra Performance Liquid Chromatography (UPLC) (ExionLC™ AD, https://sciex.com.cn/) and Tandem Mass Spectrometry (MS/MS) (QTRAP^®^6500+, https://sciex.com.cn/). The liquid chromatography conditions include the following: (1) Waters ACQUITY UPLC HSS T3 C18 column (1.8 µm, 100 mm × 2.1 mm i.d.); (2) Mobile phase: A phase, ultra-pure water (with 0.04% acetic acid); B phase, acetonitrile (with 0.04% acetic acid); (3) Gradient elution programme: 0 min A/B at 95:5 (*V*/*V*), 1.0 min A/B at 95:5 (*V*/*V*), 8.0 min at 5:95 (*V*/*V*), 9.0 min at 5:95 (*V*/*V*), 9.1 min at 95:5 (*V*/*V*), 12.0 min at 95:5 (*V*/*V*); (4) Flow rate of 0.35 mL/min; column temperature at 40 °C; injection volume of 2 μL. The mass spectrometry conditions include Electrospray Ionisation (ESI) source temperature at 550 °C, mass spectrometry voltage at 5500 V in positive ion mode, and −4500 V in negative ion mode, with curtain gas (CUR) at 35 psi. Each ion pair in Q-Trap 6500+ was detected based on the optimised declustering potential (DP) and collision energy (CE).

### 4.6. Transcriptome Sequencing and Data Analysis

Total RNA extraction, library construction, and RNA sequencing (using the Illumina NovaSeq6000/HiSeq X Ten platform) were performed by Pisenno Biotechnology Co., Ltd. (Shanghai, China). The tomato reference genome and annotation files were obtained from https://solgenomics.net/. RNA-Seq analysis was performed using the DESeq2 R package. Gene ontology (GO) enrichment analysis, Kyoto encyclopaedia of genes and genomes (KEGG) enrichment analysis, and heatmap construction were performed using ClusterProfiler (4.14.6), Rtsne (https://www.rdocumentation.org/), and pheatmap R (https://www.rdocumentation.org/) software packages.

### 4.7. Construction of CRISPR/Cas9 Vectors

Following the method described by Rui [38], the exon region of MYC2 was targeted. Guide RNA (sgRNA) was designed using the online tool CRISPR-Pv2.0 (http://cbi.hzau.edu.cn/CRISPR2/ (accessed on 28 April 2024)), and target primers containing BsaI restriction sites were synthesised and ligated into an expression vector containing Cas9 to obtain the CRISPR/Cas9-SIMYC2 recombinant plasmid. The primers are listed in Appendix A. After sequencing confirmed the successful construction of the recombinant vector, agrobacterium-mediated genetic transformation was performed on Micro-Tom tomato plants. Two homozygous transgenic lines were obtained through selection on hygromycin-resistant plates and subsequent PCR amplification and sequencing verification.

### 4.8. Construction of Overexpression Vectors

The coding sequence (CDS) of MYC2 was recombined with the pCMBIA-1300-GFP vector. The recombinant plasmid (p1300-35SN-GFP-MYC2) was transformed into agrobacterium tumefaciens. The primers are listed in Appendix A. Tomato (M82) leaf segments were used as explants, with petioles and leaf tips removed, and 0.5 cm × 0.5 cm pieces were cut for genetic transformation. Heterologous expression in tomatoes was conducted as described by Li [39]. Two *MYC2* overexpression lines were obtained and their *MYC2* transcription levels were assessed using RT-PCR.

### 4.9. Subcellular Localization of MYC2

The CDS of SlMYC2 was amplified and cloned into a pCMBIA -1300-GFP vector. Empty pCMBIA -1300-GFP and SlMYC2-pCMBIA -1300-GFP plasmids were introduced separately into GV3101 agrobacterium and transiently transformed into tobacco leaves. The primers are listed in Appendix A. After 60 h, imaging was performed by confocal laser microscopy.

### 4.10. Yeast Two-Hybrid Assay (Y2H)

The coding sequence (CDS) of SlMYC2 was amplified and cloned into a pGADT7 vector. The CDS of SlJAZ2, SlERF4, and SlERF5 were amplified and inserted into the pGBKT7 vector. The primers are listed in Appendix A. All plasmids were co-transformed into Y2H yeast strains. The transformed cells were cultured on SD/-Trp/-Leu medium at 30 °C for 2 days. Subsequently, yeast cells were selected and cultured on SD/-Trp/-Leu/-His/-Ade medium at 30 °C for 3 days to observe growth variations. Empty MYC2-pGADT7 and empty pGBKT7 vectors were used as controls.

### 4.11. Luciferase Complementation Imaging (LCI)

The CDS of SlMYC2 was amplified and cloned into an Nluc vector. The CDS of SlJAZ2, SlERF4, and SlERF5 were amplified and inserted into the Cluc vector. After successful sequencing of the MYC2-Nluc, JAZ2-Cluc, ERF4-Cluc, and ERF5-Cluc recombinant vectors, The primers are listed in Appendix A. They were introduced into GV3101 agrobacterium and transiently transformed into tobacco leaves. After 60 h, the cells were observed using an in vivo fluorescence detector. Fluorescence observation was carried out by using Living Plant Fluorescence Detector (Newton 7.0, Vilber Bio Imaging, Paris, France) on the public platform of School of Life Sciences, Qingdao Agricultural University, with specific reference to the method of Sheng [40].

### 4.12. Statistical Analysis

Statistical significance was determined using the SPSS statistical package (version 18.0; SPSS Inc., Chicago, IL, USA), with significance levels set at * *p* < 0.05, ** *p* < 0.01 (Student’s *t*-test).

## Figures and Tables

**Figure 1 plants-14-01353-f001:**
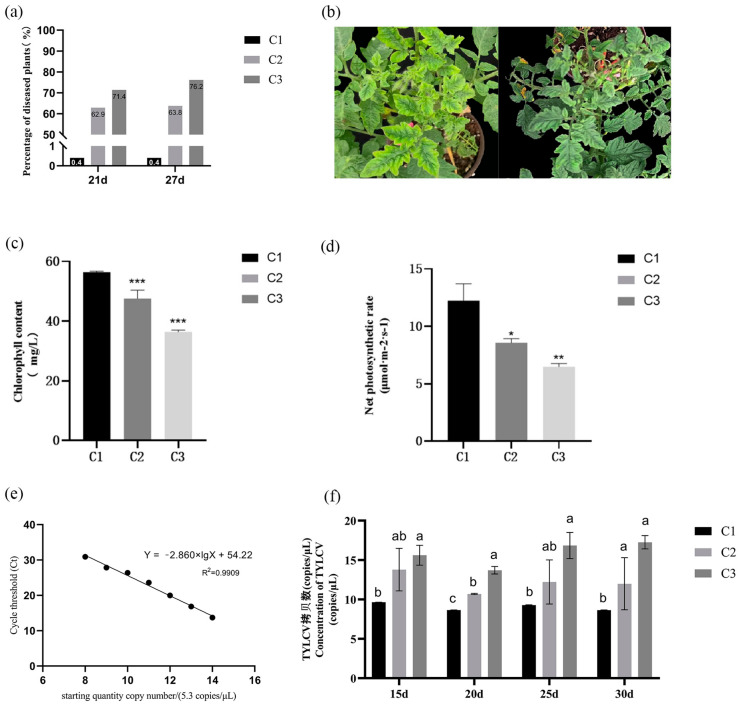
Exogenous application of CCC enhances tomato resistance to yellow curl virus disease. C1, C2, and C3, respectively, indicate the following: CCC treatment group every five days after the virus inoculation, CCC treatment every five days after the onset of the disease, and vaccination only virus-treated groups. (**a**) Incidence rates of the three treatments at two time-points. (**b**) Leaf colour significantly recovered after CCC treatment following disease onset. (**c**) Chlorophyll content on day 21 for the three treatments. (**d**) Net photosynthetic rate on day 21 for the three treatments. (**e**) Standard curve plotted based on plasmid copy number. (**f**) Viral copy number from day 15 to day 30 for the three treatments. *, **, *** and different lowercase letters indicate significant differences between treatments (*p* < 0.05).

**Figure 2 plants-14-01353-f002:**
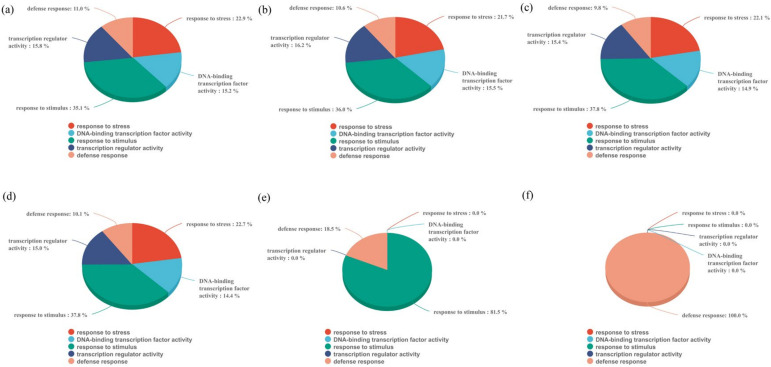
Biological processes (Gene Ontology) of differentially expressed genes (DEGs) from various comparisons. DEGs in the GO enrichment analysis for the comparisons of 0 h vs. 2 h (**a**), 0 h vs. 4 h (**b**), 0 h vs. 6 h (**c**), 0 h vs. 8 h (**d**), 0 h vs. 12 h (**e**), and 0 h vs. 24 h (**f**), respectively.

**Figure 3 plants-14-01353-f003:**
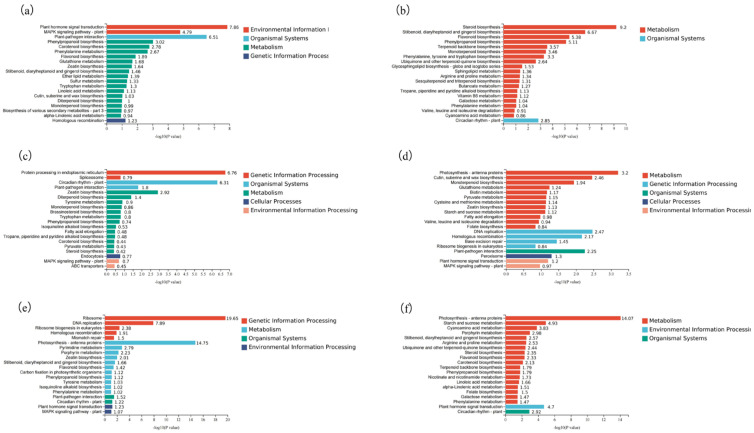
Kyoto encyclopaedia of genes and genomes (KEGG) pathways of differentially expressed genes (DEGs) from various comparisons. The KEGG pathways enriched by DEGs in the six groups of 0 h vs. 2 h (**a**), 0 h vs. 4 h (**b**), 0 h vs. 6 h (**c**), 0 h vs. 8 h (**d**), 0 h vs. 12 h (**e**), and 0 h vs. 24 h (**f**), respectively.

**Figure 4 plants-14-01353-f004:**
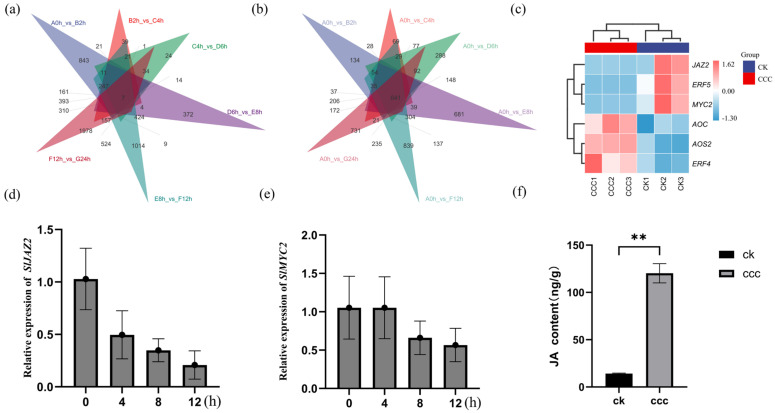
Venn diagram analysis of DEGs and the expression levels of *MYC2*, *JAZ2*, and the content JA in plants after CCC treatment. Venn diagram analysis of six periods (**a**,**b**); the cluster analysis (**c**). Gene expression levels of *MYC2* (**d**) and *JAZ2* (**e**) following CCC treatment; changes in JA content in tomato leaves after 4 h of CCC treatment (**f**). ** indicate significant differences between treatments (*p* < 0.05).

**Figure 5 plants-14-01353-f005:**
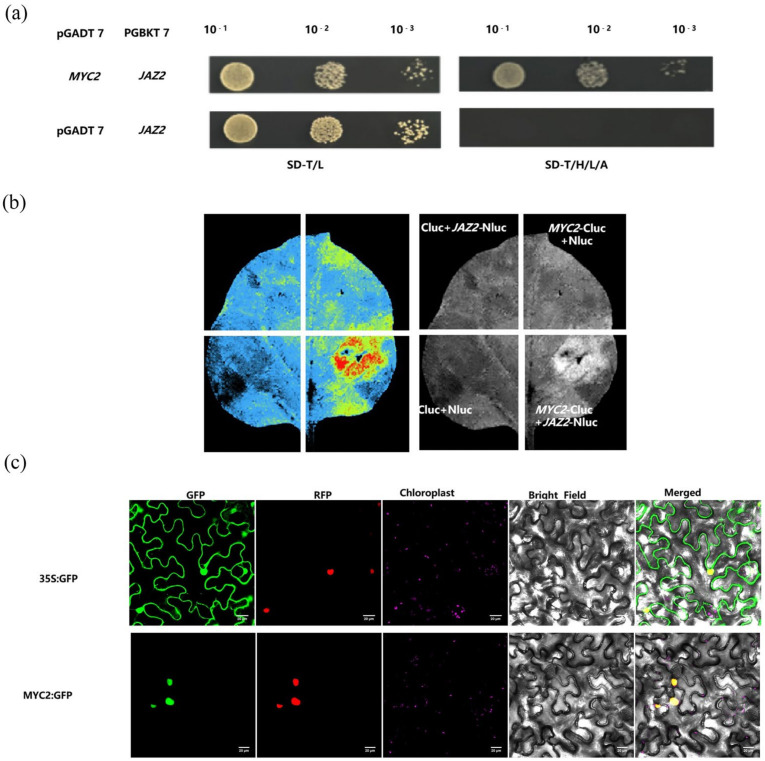
Verification of the interaction between MYC2 and JAZ2 and the subcellular localization of MYC2. Yeast two-hybrid (Y2H) and LUC assays confirmed the interaction between MYC2 and JAZ2 (**a**,**b**); Subcellular localization of MYC2 (**c**).

**Figure 6 plants-14-01353-f006:**
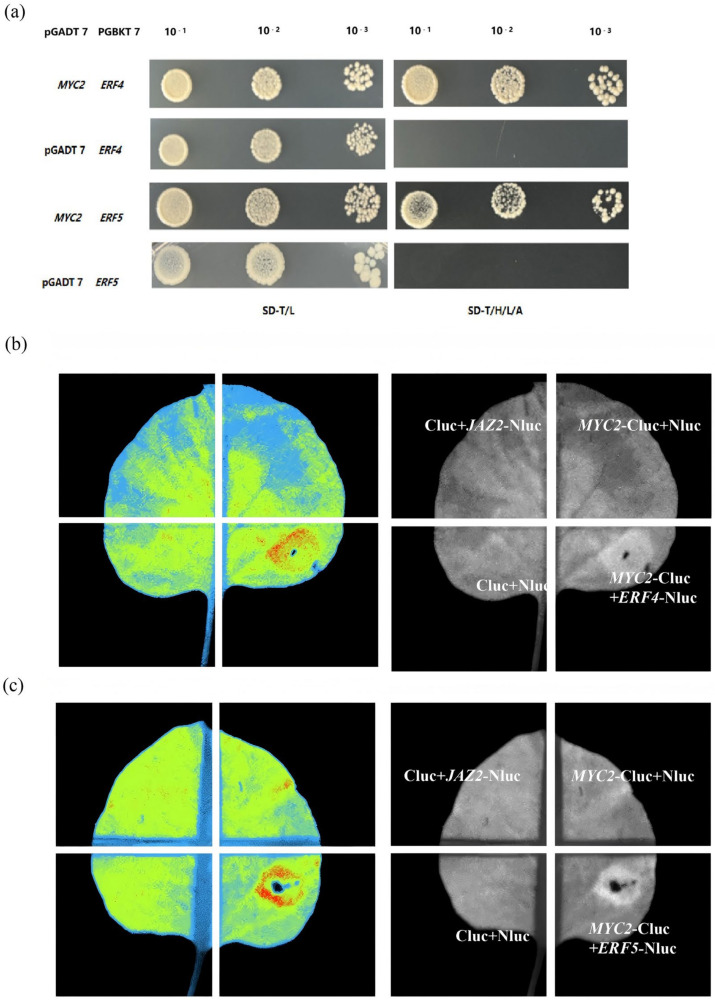
Validation of the interaction between MYC2 and ERF4, ERF5 through yeast two-hybrid assays (**a**) and luciferase complementation imaging (**b**,**c**).

**Figure 7 plants-14-01353-f007:**
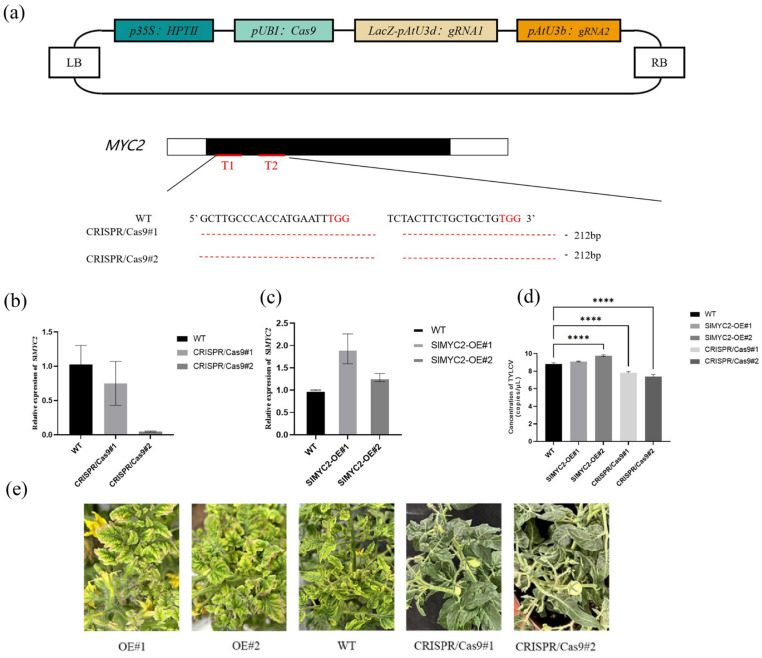
Transgenic validation of the relationship between *MYC2* and TYLCD resistance in tomatoes. Schematic representation of the construction of the CRISPR/Cas9 vector and diagram of the knockout strains (**a**); expression levels of MYC2 in the knockout lines (**b**) and overexpression lines (**c**); detection of viral copy numbers in transgenic strains following pathogen inoculation (**d**); phenotypic comparison of transgenic lines inoculated with TYLCV (**e**). **** indicate significant differences between treatments (*p* < 0.05).

**Figure 8 plants-14-01353-f008:**
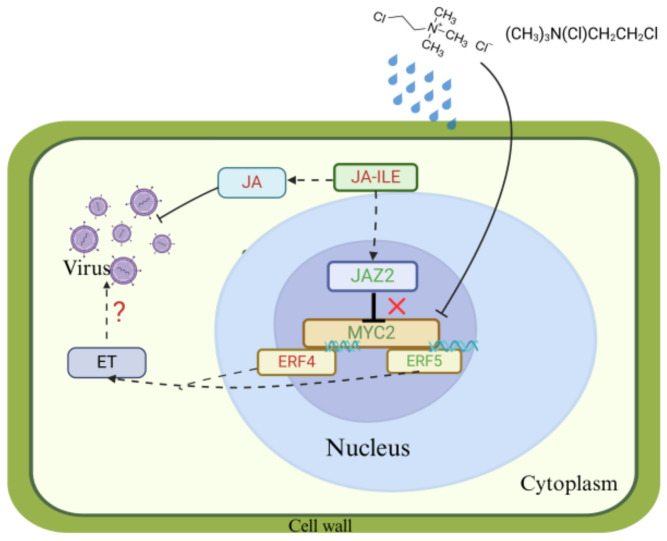
Molecular mechanism model of exogenous CCC-induced resistance to TYLCD in tomatoe. The red font indicates up-regulated expression genes, and the green font indicates down-regulated expression genes. “X” indicates that spraying CCC may break the inhibitory effect of *JAZ2* on *MYC2*.

## Data Availability

The data of this article will be available by the authors on request.

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
