# Peer review of "SlMYC2 Mediates the JA Pathway by Responding to Chlorocholine Chloride in the Regulation of Resistance to TYLCD"

_plants, 2025, doi:10.3390/plants14091353_

Round 1
Reviewer 1 Report
Comments and Suggestions for Authors
In this study, the authors carried out to prove the hypothesis through various experiments, and I think this is a good paper overall. However, there are some issues that need to be improved.
4.1 Line 344-348
The authors should describe the growing conditions for tomatoes in more detail, including light conditions, temperatures and humidities, pot size, fertilizing condition, so forth.
Line 355
I think you are referring to infectious clones, but please describe clearly what TY-1 and TY-2 represent.
Line 360-362
Please describe how to obtain CCC, including product name. Please also describe mol/L so that the amount of active ingredient can be understood, not just mg/L.
Line 452-458
The explanation of experimental methods are too simplified and difficult to understand. Please cite a paper or provide more detailed information, such as how luciferin emits light, so forth.
Fig7
The TYLCV titer in MYC2 knockout tomato plants is at most 80% of that of the wild type. Moreover, although yellowing symptoms are alleviated, the leaflets are severely malformed in these knockout plants (Fig7 e). Please check that the unit of the vertical axis showing the TYLCV titer are correct. (Fig7 d)
Line 338-340, Fig 8
In JA-stimulated cells, the physical interaction between JAZ proteins and transcriptional activators such as MYCs is broken, which results in derepression of the JA signaling pathway and activation of a large number of JA-responsive gene such as plant defense genes. In the present study, the authors clarified that application of CCC to tomato plants have increased the activity of various defense-related genes, which resulted in suppression of TYLCV proliferation and alleviation of yellowing symptoms. Application of CCC has increased JA, and has decreased the transcriptional activity of MYC2 and JAZ2 genes, with increasing AOC transcriptional activity. CCC application also altered ERF gene activity. The authors also found that MYC2 protein interacted or bound to JAZ2 and ERF proteins. All that can be deduced from these results is that CCC may increase tomato plant resistance against TYLCV by affecting the JA and/or ET signaling pathways, accompanied by a change (decrease) in the activity of the MYC2 gene. I think this is just only conclusion the authors can maintain in this study. I don't think it can be insisted that the decrease in activity of the MYC2 gene would control the mode of action of CCC as a key enzyme that regulates the JA transduction pathway. Therefore, I think the model shown in Fig. 8 is too much of an exaggeration and unnecessary. I recommend that you reconsider about this matter.
That is all.
Author Response
Comments 1:4.1 Line 344-348:The authors should describe the growing conditions for tomatoes in more detail, including light conditions, temperatures and humidities, pot size, fertilizing condition, so forth.
Response 1:According to the opinions of reviewer, we added a detailed description of tomato growth conditions in Part 4.1(line 377-383).
Comments 2:Line 355:I think you are referring to infectious clones, but please describe clearly what TY-1 and TY-2 represent.
Response 2:In section 4.1.1, we added the description of infectious clones TY-1 and TY-2 (line 385-388).
Comments 3:Line 360-362:Please describe how to obtain CCC, including product name. Please also describe mol/L so that the amount of active ingredient can be understood, not just mg/L.
Response 3:The description of CCC and its reagent preparation method has been added in 4.1.2 (line 401-405).
Comments 4:Line 452-458:The explanation of experimental methods are too simplified and difficult to understand. Please cite a paper or provide more detailed information, such as how luciferin emits light, so forth.
Response 4:We added the supplement of LCI experiment (line 507-510) and the reference (line 644-47).
Comments5:Fig7:The TYLCV titer in MYC2 knockout tomato plants is at most 80% of that of the wild type. Moreover, although yellowing symptoms are alleviated, the leaflets are severely malformed in these knockout plants (Fig7 e). Please check that the unit of the vertical axis showing the TYLCV titer are correct. (Fig7 d)
Response 5:The knockout tomato plants leaflets showed severely malformed. We guessed that it might be caused by the loss of function of SlMYC2. The CDS region of the knockout tomato plants were missing a large sequence with a length of 212bp, and SlMYC2 is a very important transcription factor in plants, which can regulate multiple growth and development-related pathways. The loss of SlMYC2 function may have caused abnormal leaves of tomato. And Fig. 7d shows the copy number of the virus, so the unit of the vertical axis is orrect.
Comments 6:Line 338-340, Fig 8:In JA-stimulated cells, the physical interaction between JAZ proteins and transcriptional activators such as MYCs is broken, which results in derepression of the JA signaling pathway and activation of a large number of JA-responsive gene such as plant defense genes. In the present study, the authors clarified that application of CCC to tomato plants have increased the activity of various defense-related genes, which resulted in suppression of TYLCV proliferation and alleviation of yellowing symptoms. Application of CCC has increased JA, and has decreased the transcriptional activity of MYC2 and JAZ2 genes, with increasing AOC transcriptional activity. CCC application also altered ERF gene activity. The authors also found that MYC2 protein interacted or bound to JAZ2 and ERF proteins. All that can be deduced from these results is that CCC may increase tomato plant resistance against TYLCV by affecting the JA and/or ET signaling pathways, accompanied by a change (decrease) in the activity of the MYC2 gene. I think this is just only conclusion the authors can maintain in this study. I don't think it can be insisted that the decrease in activity of the MYC2 gene would control the mode of action of CCC as a key enzyme that regulates the JA transduction pathway. Therefore, I think the model shown in Fig. 8 is too much of an exaggeration and unnecessary. I recommend that you reconsider about this matter.
Response 6:In previous reports, JZA2 is located upstream of MYC2, and JAZ2 inhibits the expression of MYC2 in tomato, so MYC2 should be up-regulated when JAZ2 is down-regulated. However, in our study, after CCC treatment, both JAZ2 and MYC2 in tomatoes were down-regulated, so we want to present our hypothesis in Figure 8: that is to say, exogenous CCC may break the inhibitory effect of JAZ2 on MYC2. As negative regulatory factors of JA pathway, both of them are down-regulated after spraying CCC, which leads to the increase of JA content in tomatoes and ultimately improves the systemic resistance of plants. If the reviewer thinks this pattern picture is not suitable, we can consider deleting it from the manuscript.
Reviewer 2 Report
Comments and Suggestions for Authors
Please consider the following comments and revise the manuscript accordingly-
- The title is quite long, and the phrase 'Tomato Yellow Leaf Curl Disease in Tomato' is repetitive. Please revise the title to make it more concise.
- The abstract clearly states the study’s objective; however, it should also clearly present the hypothesis, such as "CCC enhances TYLCD resistance by modulating the JA pathway." Additionally, the abstract should mention how these findings can be applied in agricultural practices and how this research could benefit tomato farmers or commercial crop production.
- The introduction section contains long, dense sentences. You can break down complex ideas into simpler, more digestible segments. Some points about JA pathways and MYC2’s role are repeated multiple times. Please condense these repetitions. The introduction section provides strong background information but does not state the knowledge gap. You should add a paragraph which focus on the study aim rather than summarizing findings.
- “2.1. The Effect of Exogenous Application of CCC on Tomato Resistance to TYLCD” section mentions the connection to physiological pathways. If possible, please add a discussion on the molecular mechanisms would add depth.
- In Lines 112-113 mention the phrase "treatment group one, the incidence of disease in the plants was notably lower than in the other three groups" which is unclear. Please Consider specifying the exact treatment groups in a structured way.
- In “2.2. GO Enrichment Analysis of CCC Treatment”, you mention significance of some GO terms in relation to CCC’s role in disease resistance which could be expanded. In addition, please provide a more detailed discussion on how specific DEGs contribute to tomato resistance.
- The numbering of the headings in “3.3. KEGG Enrichment Analysis of DEGs Processed by CCC” section is not sequential; you used 3.3 instead of 2.3. Please make it correct.
- In “3.3. KEGG Enrichment Analysis of DEGs Processed by CCC” section, some pathways are only briefly mentioned; you can provide a more in-depth analysis of key pathways (e.g., plant hormone signal transduction).
- In “2.6. Knockout and Overexpression of MYC2 Affect the Disease Resistance of Tomatoes” section, you can provide more details on how MYC2 overexpression negatively impacts resistance. In addition, you can mention the potential applications of MYC2 manipulation in agricultural disease management.
- “3.1. Exogenous Application of CCC Can Effectively Enhance Resistance to TYLCD” section should begin with a clearer hypothesis about how CCC influences TYLCD resistance. Some points are repeated in this section, such as the mention of CCC reducing TYLCD symptoms and improving photosynthesis. Please merge these. This section briefly mentions that CCC could be a practical alternative to genetic modifications. However, you can add a concluding sentence about its feasibility in large-scale farming would strengthen its impact.
- “3.2. SlMYC2 Mediates JA and ET Pathways by Responding to CCC in the Regulation of Resistance to TYLCD in Tomato” section, you mention that MYC2 is downregulated after CCC treatment, which differs from existing research. Please consider discussing why this might be happening (e.g., possible feedback inhibition or post-translational modifications) instead of just stating the difference.
- “4.1. Growth Conditions and Treatment of Experimental Materials” section, you can add the specific environmental parameters (temperature, humidity, light intensity) for better reproducibility, and please mention how plants were monitored for uniform growth would help clarify plant selection criteria.
- In “4.1.1. Infectious Clone Inoculation” section, you should mention a control group for inoculation (mock-inoculated plants)
- In “4.1.2. CCC Treatment” section, you mention 17:00- 18:00 as the spraying time. Why was 17:00–18:00 chosen as the spraying time? Some justification (e.g., to minimize evaporation) would be helpful.
- In “4.2. Incidence Rate Statistics” section you should mention how many replicates (e.g., number of independent trials) were used.
- In “4.3. Measurement of Chlorophyll Content and Net Photosynthetic Rate” section, if possible, please include whether SPAD readings were validated with chemical chlorophyll extraction.
Author Response
Comments 1: The title is quite long, and the phrase 'Tomato Yellow Leaf Curl Disease in Tomato' is repetitive. Please revise the title to make it more concise.
Response 1:We simplified the title of the article (line 2-3).
Comments 2: The abstract clearly states the study’s objective; however, it should also clearly present the hypothesis, such as "CCC enhances TYLCD resistance by modulating the JA pathway." Additionally, the abstract should mention how these findings can be applied in agricultural practices and how this research could benefit tomato farmers or commercial crop production.
Response 2: In the abstract, we have added the hypothesis and application about the molecular mechanism of CCC improving tomato disease resistance (line 29-34).
Comments 3: The introduction section contains long, dense sentences. You can break down complex ideas into simpler, more digestible segments. Some points about JA pathways and MYC2’s role are repeated multiple times. Please condense these repetitions. The introduction section provides strong background information but does not state the knowledge gap. You should add a paragraph which focus on the study aim rather than summarizing findings.
Comments 3:We have revised some words in the introduction, and simplified the viewpoints related to MYC2 and JA pathways (line 42-81). In addition, in the last paragraph of the introduction, we added a description of the knowledge gap and the research purpose of this study (line 94-102).
Comments 4: “2.1. The Effect of Exogenous Application of CCC on Tomato Resistance to TYLCD” section mentions the connection to physiological pathways. If possible, please add a discussion on the molecular mechanisms would add depth.
Comments 4:We added this part to the discussion section (line 311-319).
Comments 5: In Lines 112-113 mention the phrase "treatment group one, the incidence of disease in the plants was notably lower than in the other three groups" which is unclear. Please Consider specifying the exact treatment groups in a structured way.
Comments 5:We have re-described this part of the text (line 116-119).
Comments 6: In “2.2. GO Enrichment Analysis of CCC Treatment”, you mention significance of some GO terms in relation to CCC’s role in disease resistance which could be expanded. In addition, please provide a more detailed discussion on how specific DEGs contribute to tomato resistance.
Comments 6:We found that the JA content in plants increased significantly after CCC spraying, so we paid special attention to the genes related to JA signaling pathway in the analysis results of transcriptome data. Fortunately, the expression trend of JAZ2 and MYC2 showed a continuous downw-regulated trend at different times after CCC treatment, so the content of this manuscript was obtained. Because our experiment does not involve other DEGs, we think that their matching degree with our manuscript is not very high, and no further discussion has been carried out.
Comments 7: The numbering of the headings in “3.3. KEGG Enrichment Analysis of DEGs Processed by CCC” section is not sequential; you used 3.3 instead of 2.3. Please make it correct.
Comments 7:We have corrected the headings number (line 166).
Comments 8: In “3.3. KEGG Enrichment Analysis of DEGs Processed by CCC” section, some pathways are only briefly mentioned; you can provide a more in-depth analysis of key pathways (e.g., plant hormone signal transduction).
Comments 8:Through the personalized analysis of DEGs, it is found that DEGs with the same expression trend in different periods after CCC treatment is only found in JA and ET pathways, and the above two pathways are also discussed in detail in the discussion part 3.2.
Comments 9: In “2.6. Knockout and Overexpression of MYC2 Affect the Disease Resistance of Tomatoes” section, you can provide more details on how MYC2 overexpression negatively impacts resistance. In addition, you can mention the potential applications of MYC2 manipulation in agricultural disease management.
Response 9: We have added a detailed description (line 255-265).
Comments 10: “3.1. Exogenous Application of CCC Can Effectively Enhance Resistance to TYLCD” section should begin with a clearer hypothesis about how CCC influences TYLCD resistance. Some points are repeated in this section, such as the mention of CCC reducing TYLCD symptoms and improving photosynthesis. Please merge these. This section briefly mentions that CCC could be a practical alternative to genetic modifications. However, you can add a concluding sentence about its feasibility in large-scale farming would strengthen its impact.
Response 10: We have made corresponding modifications and improvements in Part 3.1 (line 311-325).
Comments 11:“3.2. SlMYC2 Mediates JA and ET Pathways by Responding to CCC in the Regulation of Resistance to TYLCD in Tomato” section, you mention that MYC2 is downregulated after CCC treatment, which differs from existing research. Please consider discussing why this might be happening (e.g., possible feedback inhibition or post-translational modifications) instead of just stating the difference.
Response 11: What we mentioned in this paper is different from other people's research results, which means that JAZ2 is an inhibitor of MYC2 in other people's research results, so the expression levels of JAZ2 and MYC2 in tomato should be opposite. However, in our results, the expression levels of JAZ2 and MYC2 decreased at the same time after exogenous CCC spraying, and we explained this reason in the following content (line 352-357).
Comments 12: “4.1. Growth Conditions and Treatment of Experimental Materials” section, you can add the specific environmental parameters (temperature, humidity, light intensity) for better reproducibility, and please mention how plants were monitored for uniform growth would help clarify plant selection criteria.
Response 12: We have added a detailed description of tomato growth conditions in Part 4.1(line 377-383).
Comments 13: In “4.1.1. Infectious Clone Inoculation” section, you should mention a control group for inoculation (mock-inoculated plants)
Response 13: The treatment method of the control group has been supplemented (line 398-399).
Comments 14: In “4.1.2. CCC Treatment” section, you mention 17:00- 18:00 as the spraying time. Why was 17:00–18:00 chosen as the spraying time? Some justification (e.g., to minimize evaporation) would be helpful.
Response 14: We explained it in the article (line 414-417).
Comments 15: In “4.2. Incidence Rate Statistics” section you should mention how many replicates (e.g., number of independent trials) were used.
Response 15: As mentioned in the first sentence of section 4.2, the number of plants in each group is 35.
Comments 16: In “4.3. Measurement of Chlorophyll Content and Net Photosynthetic Rate” section, if possible, please include whether SPAD readings were validated with chemical chlorophyll extraction.
Response 16:We have tested the chlorophyll data measured by SPAD by chemical methods, and the trend of the two is the same. The following figure shows the results of chemical methods.
Round 2
Reviewer 1 Report
Comments and Suggestions for Authors
The all parts I pointed out previously have been properly corrected. Therefore, I recommend that this paper would be published in journal of Plants.
Reviewer 2 Report
Comments and Suggestions for Authors
The suggested revision has improved the quality of the manuscript.